# Pain Chronicity and Relief: From Molecular Basis to Exercise-Based Rehabilitation

**DOI:** 10.3390/biology14091116

**Published:** 2025-08-23

**Authors:** Weidi Ni, Xin Kuang, Zheng Zhu

**Affiliations:** School of Exercise and Health, Shanghai University of Sport, Shanghai 200438, China; 2321518037@sus.edu.cn (W.N.); 2421518038@sus.edu.cn (X.K.)

**Keywords:** chronic pain, exercise, analgesia, central mechanisms, peripheral mechanisms

## Abstract

Millions of people struggle with chronic pain, a condition where pain lingers for over three months, and pharmacological treatments are not always a safe or effective solution. In this review, we conduct a comprehensive examination of the scientific evidence surrounding the mechanisms of chronic pain and to elucidate the physiological processes through which exercise alleviates pain. The results confirm that exercise prompts the brain to release its own natural painkillers, which helps to block pain signals. In addition, it calms down the nervous system, which often becomes hypersensitive and overreactive, causing pain to persist. At the same time, exercise helps reduce inflammation at the site of pain. This research confirms that exercise is a true biological treatment that helps reverse the underlying causes of chronic pain. By embracing exercise, people living with chronic pain can take an active role in their recovery, reduce their suffering, and reclaim a better quality of life.

## 1. Introduction

With the International Association for the Study of Pain (IASP) redefining chronic pain as a disease where pain persists or recurs for more than three months, surpassing the usual tissue healing period, global attention to pain management has intensified [1]. While acute pain is a dynamic healing process in response to tissue trauma, once the acute injury has healed, if pain persists, its original protective function diminishes, instead becoming a persistent physical and psychological burden. Studies estimate that approximately 20.9% of U.S. adults experience chronic pain, with high-impact chronic pain affecting nearly 7% of individuals, severely restricting daily activities [2]. Chronic pain is usually comorbid with other disorders such as anxiety, depression and insomnia, imposing a substantial socioeconomic burden due to increased healthcare expenditures, reduced workforce productivity, and significant disability burdens [3,4]. Notably, these factors also predispose individuals to chronic pain. As a leading cause of disability, chronic pain not only leads to diminished quality of life but also increases mortality risks, particularly due to its association with cardiovascular diseases [5]. Thus, addressing chronic pain is not only a matter of improving individual health outcomes but also a critical public health priority that demands coordinated action from healthcare providers, policymakers, and researchers.

Clinically, chronic pain commonly occurs in the head, shoulders, lower back, and knees; it can be categorized into three primary types: nociceptive, neuropathic, and nociplastic [6]. Nociceptive pain arises from tissue damage or inflammation, neuropathic pain stems from nerve injury or dysfunction such as diabetic neuropathy, and nociplastic pain results from central nervous system sensitization without clear tissue or nerve damage such as chronic low back pain. However, due to its heterogeneous clinical presentations, chronic pain often resists accurate diagnosis and optimal management.

Understanding the molecular mechanisms underlying chronic pain is crucial for the development of effective management strategies. A large body of research has detailed the roles of aberrant neuroplasticity, sustained immune activation, and epigenetic modulation in pain chronicity [7,8,9,10]. Building upon this foundation of knowledge, current research is delving deeper into this topic, focusing on revealing immune cell heterogeneity [11], sex-specific responses [12], and dynamic alterations in brain connectivity networks [13]. Despite leading to significant advances, this knowledge has primarily driven pharmacological development. However, medications such as non-steroidal anti-inflammatory drugs, opioids, and neuropathic pain agents such as gabapentinoids may come with severe side effects and risk of addiction, limiting their long-term use [14]. Meanwhile, exercise-based rehabilitation has emerged as a highly effective, accessible, and cost-effective non-pharmacological intervention for a variety of chronic pain conditions. However, the molecular mechanisms by which exercise exerts its analgesic effects are often discussed separately from the molecular pathophysiology of chronic pain itself. This gap in the literature represents a key focus of our review. While many studies cover the molecular basis of pain and others the benefits of exercise, few systematically connect the two.

Our review aims to bridge this gap by synthesizing current knowledge on the molecular pathways of chronic pain with the known mechanisms of exercise rehabilitation. By directly comparing the pathological molecular cascades of chronic pain with the corrective adaptations induced by exercise, we provide a robust biological rationale for its use and lay the groundwork for developing targeted, mechanism-based rehabilitation strategies. This integrated perspective is essential for moving beyond general exercise recommendations toward the development of more rational, targeted, and personalized exercise prescriptions for individuals with chronic pain.

## 2. Search Strategy

Relevant studies were identified by searching academic databases, including PubMed, Web of Science, and Google Scholar, performed in July 2025. The search strategy was designed to be broad and inclusive, capturing literature on three core concepts: (1) chronic pain, (2) exercise-based intervention, and (3) the biological mechanisms. The search terms applied to all fields (title, abstract, and text word), combined using Boolean operators (AND, OR), are as follows: ‘chronic pain’ OR ‘nociceptive pain’ OR ‘neuropathic pain’ OR ‘nociplastic pain’ OR ‘central sensitization’ OR ‘peripheral sensitization’ OR ‘analgesia’ OR ‘hypoalgesia’ AND ‘mechanism’ OR ‘molecular’ OR ‘cellular’ OR ‘pathway’ OR ‘neurobiology’ AND ‘training’ OR ‘exercise’ OR ‘physical activity’ OR ‘physical therapy’ OR ‘rehabilitation’ OR ‘exercise-induced analgesia’ OR ‘exercise-induced hypoalgesia’. An additional search was carried out whereby we examined the references of the included articles.

To be included, a study needed to meet the following criteria: (a) subjects: both human clinical studies and preclinical animal models were included to provide a broad, translational perspective on the topic; (b) pain condition: no restrictions were placed on the specific type of chronic pain; (c) intervention: all forms of exercise were considered; (d) publication type: original research articles and review articles were included; (e) language: the search was limited to original research written in English. Case reports, letters to the editor, conference abstracts, and non-peer-reviewed articles were not eligible for inclusion.

## 3. Classification and Mechanisms of Chronic Pain

Chronic pain arises from complex and interrelated mechanisms that evolve following tissue injury, infection, or prolonged inflammation. Several classification systems for this condition have been proposed, often based on its etiology, anatomical location, or predominant underlying mechanisms. A common approach categorizes chronic pain into three major types based on its likely origin: nociceptive, neuropathic, or nociplastic pain. Alongside these three major types, the mixed pain phenotype is increasingly recognized by clinicians and researchers (Figure 1). While different pain classifications may have distinct initiating factors, the processes leading to pain chronicity converge on fundamental mechanisms, which can be broadly categorized into peripheral and central mechanisms (Figure 2). This section outlines the primary classifications of chronic pain, followed by a comprehensive discussion of the general pathogenetic mechanisms that drive chronicity, focusing on overarching peripheral and central processes.

### 3.1. Classification of Chronic Pain

Nociceptive pain originates from direct non-neural tissue damage and inflammation. Clinically, it is categorized based on its origin, as somatic pain, arising from tissues such as skin, muscles, and bones, or visceral pain, which originates from internal organs [15]. In nociceptive pain, the release of inflammatory mediators leads to the activation and sensitization of peripheral nociceptors, resulting in the robust and direct transmission of pain signals to the central nervous system (CNS) [16]. The subsequent peripheral sensitization is characterized by the upregulation of ion channel activity and an altered profile of receptor expression, which together contribute to a persistent state of hyperexcitability.

In contrast, neuropathic pain is primarily a consequence of nerve injury or disease. Following nerve damage, there is a dysregulation of ion channels—particularly sodium channels such as Nav1.7 and Nav1.8—become dysregulated, which leads to ectopic discharges and spontaneous neuronal firing [17]. This aberrant electrical activity is compounded by maladaptive plastic changes in the CNS, including the reorganization of synaptic connections and the activation of glial cells. These changes not only perpetuate the pain state but also render neuropathic pain notoriously resistant to conventional analgesics.

Nociplastic pain, a concept that has gained recognition in recent years, is defined by the presence of altered central pain processing in the absence of obvious peripheral tissue or nerve damage [18]. The introduction of this term represents a critical advancement in this field, as it provides a distinct mechanistic descriptor for chronic pain syndromes, such as fibromyalgia, that were previously considered idiopathic [19]. Functional neuroimaging and electrophysiological studies have revealed that in brain regions integral to pain perception, such as the anterior cingulate cortex (ACC), insula, and prefrontal cortex, abnormal connectivity and dysregulated neurotransmitter activity occur in nociplastic pain states [20]. This results in the amplification of pain signals due to an imbalance between excitatory (glutamatergic) and inhibitory (GABAergic) neurotransmission, compounded by dysfunction of descending pain modulation pathways. In addition, genetic predisposition and psychological factors such as stress, anxiety, and depression can influence the development and maintenance of nociplastic pain [21].

In clinical practice, many patients do not present with a single, isolated type of pain but rather exhibit a complex interplay of these mechanisms. The concept of “mixed pain”, representing interactions and overlaps between different pain mechanisms, including nociceptive, neuropathic, and nociplastic components, is increasingly recognized, though its precise definition and diagnostic criteria are still evolving [22].

An initial nociceptive insult, typically resulting from tissue injury or inflammation, may serve as a trigger for peripheral sensitization. When such stimuli persist, they can eventually lead to nerve damage, thereby introducing neuropathic components. Simultaneously, the ongoing barrage of nociceptive input drives central sensitization, creating an environment in which the normally distinct pathways of pain processing begin to overlap. Over time, the convergence of peripheral and central mechanisms gives rise to a mixed pain phenotype, characterized by simultaneous alterations in peripheral ion channel function, maladaptive central plasticity, and impaired descending inhibitory controls. The recognition that chronic pain is not a uniform entity but rather a heterogeneous condition with overlapping molecular mechanisms has profound implications for both research and clinical practice, presenting a significant challenge that necessitates a fundamental shift from unimodal, mechanism-specific interventions towards more holistic, personalized treatment strategies capable of addressing the multifaceted nature of pain experience [23].

### 3.2. Peripheral Mechanisms of Chronic Pain

Peripheral tissue damage, persistent inflammation, and nerve injury triggers a cascade of biochemical events resulting from damage to tissues and surrounding cells, including cytokines such as interleukin-6 (IL-6), interleukin-1β (IL-1β) and tumor necrosis factor-α (TNF-α); chemokines; prostaglandins; bradykinin; neuropeptides; and neurotrophic factors such as nerve growth factor (NGF) and brain-derived neurotrophic factor (BDNF) [24,25]. These mediators interact with receptors on the surface of peripheral nociceptors and are responsible for initiating intracellular signaling cascades [26], involving protein kinases such as protein kinase A, protein kinase C, and mitogen-activated protein kinases (MAPK) [27,28], which may result in the modulation of ion channels that are critical for pain transmission. Among these, transient receptor potential (TRP) channels (such as TRPV1 and TRPA1), voltage-gated sodium channels (notably Nav1.7 and Nav1.8), and various types of voltage-gated calcium channels on primary afferent neurons and their somata in the dorsal root ganglia (DRG) have been identified as key players in the process of chronic pain [29,30,31].

Recent evidence revealed that NGF plays a prominent role in this process, binding to its high-affinity receptor TrkA on nociceptors [32]. NGF-TrkA signaling activates downstream pathways (including MAPK) that are crucial for modulating the expression and function of ion channels and receptors, significantly contributing to sensitization [33]. These signaling cascades lead to a lower activation threshold, increased responsiveness, and spontaneous firing of nociceptors, a phenomenon known as peripheral sensitization [7]. Recent studies have demonstrated that selective modulation of TRPV1 and specific MAPK isoforms can significantly reduce nociceptor hyperexcitability, thereby highlighting promising therapeutic avenues [34,35]. However, peripheral sensitization is not merely a passive response to inflammation but is actively amplified by the nervous system through a process known as neurogenic inflammation [36]. A key peripheral driver of this is the axonal reflex (AR), a purely local mechanism. When peripheral C-fibers are activated by noxious stimulation, the resulting action potential travels not only orthodromically toward the spinal cord but also antidromically along collateral branches to trigger an AR, which leads to the release of neuropeptides, primarily substance P and calcitonin gene-related peptide (CGRP), directly at the site of injury [37]. Substance P increases vascular permeability, leading to plasma extravasation and edema (swelling), while CGRP, a potent vasodilator, causes an increase in local blood flow (redness/flare). Together, they initiate a localized inflammatory response, contributing to the “inflammatory soup” and further sensitizing nearby nerve endings.

The peripheral environment also contains various non-neuronal cells that actively participate in peripheral sensitization and neuroinflammation. Beyond satellite glial cells in the DRG and macrophages which are noted in [38], mast cells, fibroblasts and Schwann cells release a spectrum of mediators, including cytokines, chemokines and growth factors (like NGF) after injury or inflammation, significantly contributing to the initiation and maintenance of peripheral sensitization through interaction with nearby nociceptive neurons [39,40]. In addition, elevated intracellular calcium levels and the generation of reactive oxygen species serve to further modulate gene expression within the affected neurons, particularly in DRG neurons, establishing long-term changes in cellular function [41,42]. The activation of key transcription factors and subsequent transcriptional reprogramming within the affected neurons may establish long-term functional changes, such as sustained hyperexcitability and altered neurotransmitter synthesis, which are crucial for consolidating the sensitized state and contributing to the transition from acute to chronic pain [43].

Therefore, what begins as a purely nociceptive–inflammatory process can, over time, induce sufficient structural damage to the peripheral nerves themselves, causing the pain to acquire a neuropathic component. This explains why conditions like severe osteoarthritis or rheumatoid arthritis can eventually present with features of neuropathic pain, such as burning sensations or allodynia.

### 3.3. Central Mechanisms of Chronic Pain

Persistent pain signals from the periphery are transmitted to the CNS, where they become further amplified in a process known as central sensitization. This occurs when spinal cord neurons, particularly in the dorsal horn, become hyper-responsive to incoming signals, leading to an exaggerated pain perception. Additionally, there is reorganization of pain-processing brain regions, such as the ACC and insular cortex, which enhances the perception of pain. This interaction between peripheral and central mechanisms creates a feedback loop that perpetuates the pain state.

A key cellular mechanism underlying chronic pain is LTP of synaptic connections between primary afferent fibers and spinal neurons [44,45]. This LTP is primarily mediated by NMDA receptor activation following persistent noxious input, leading to increased intracellular calcium and subsequent downstream signaling pathways that enhance synaptic transmission and neuronal excitability [46]. Under such conditions, primary afferent depolarization (PAD) is more likely to reach its threshold, triggering the dorsal root reflex (DRR), a centrally mediated phenomenon which actively perpetuates peripheral inflammation [47]. DRR generates antidromic action potentials that travel from the spinal cord back to the peripheral terminals of the sensory neuron, leading to the release of substance P and CGRP into the periphery [48]. This mechanism can spread and sustain neurogenic inflammation over a wider area than the initial injury, contributing to widespread pain and referred hyperalgesia [49]. This conversion of presynaptic inhibition into an excitatory, efferent signal represents a critical form of maladaptive plasticity, creating a vicious cycle linking central sensitization back to peripheral nociception.

Beyond the initial neuronal plasticity involving NMDA receptors, glial cells, including microglia and astrocytes, play critical roles in the development and maintenance of central sensitization [50]. Microglia and astrocytes, in particular, exhibit temporal differences and functional diversity during the progression of chronic pain. Initially, microglia are rapidly activated and release pro-inflammatory cytokines and chemokines, contributing to early neuronal sensitization [9,51]. As chronic pain persists, astrocytes become more prominent, undergoing morphological and functional changes via reactive astrogliosis, further releasing neuroactive mediators, thereby sustaining the long-term transmission of pain signals [9]. Recent studies have further revealed that the role of glial cells is not limited to cytokine release; they can also influence neurotransmitter release and clearance (such as glutamate transporters) [52], and modulate ion channel function (such as BDNF-mediated changes in chloride gradient) [53,54,55], thereby inducing neuronal disinhibition and further enhancing pain signal transmission. Pharmacological interventions targeting NMDA receptor activity and glial cell function have shown promise in alleviating central sensitization, offering new avenues for chronic pain management [56].

The dysregulation of descending pain modulatory pathways from the brainstem and cortex, which originate in the periaqueductal gray (PAG), rostral ventromedial medulla (RVM), and ventrolateral medulla (VLM), also contributes to central sensitization [57]. Normally, these pathways can either inhibit or facilitate pain transmission in the spinal cord [58,59]. In chronic pain, an imbalance occurs, often with enhanced descending facilitation and/or impaired inhibition, amplifying spinal pain signals. Neurotransmitter systems like the glutamatergic and serotonergic systems are key players in this process. Interestingly, the role of serotonin (5-HT) is complex: 5-HT1A, 5-HT1B, 5-HT2A, 5-HT6, and 5-HT7 receptors can mediate inhibition, while 5-HT3 receptors promote facilitation, particularly in persistent pain [60,61,62]. Antidepressants like tricyclic antidepressants and 5-HT-norepinephrine reuptake inhibitors are thought to exert analgesic effects partly through the enhancement of supraspinal descending inhibition through inhibiting the reuptake of 5-HT and increasing the concentration of this monoamine neurotransmitter in the synaptic cleft [63]. Dysfunction in these descending pathways significantly contributes to sustained pain and altered pain perception.

### 3.4. Sex Differences in Chronic Pain

Sex differences have been noted in many chronic pain conditions, with women generally reporting a higher prevalence, greater sensitivity, and longer duration across a range of conditions, including arthritis and migraine [64]. Evidence accumulated over the past decade shows that most of the molecular nodes sustaining chronic pain are sexually dimorphic. The actions of sex hormones, particularly estrogen and testosterone, have been widely investigated due to their significant contributions to this divergence. Testosterone is suggested to be anti-nociceptive and protective [65], while estrogen often demonstrates pro-nociceptive activity whereby it modulates receptors and immune cell activity [66,67,68]. Furthermore, it was discovered that prolactin selectively sensitizes the TRP channels TRPV1, TRPA1, and TRPM8 in female neurons, and causes increased excitability in nociceptors [69,70]. Another topic being considered regarding sex differences is distinct neuro-immune interactions. Male pain hypersensitivity often critically depends on spinal microglia, specifically involving Toll-like receptor 4 (TLR4) activation, while females frequently rely on adaptive immune cells like T cells and B cells for nociceptive sensitization [71,72,73]. Recent advances in large-scale transcriptomic reveal sex-specific gene expression profiles across the DRG and brain regions like the ACC and amygdala for neuropathic pain [74,75,76]. Despite significant progress in this field, translating identified differentially expressed genes into clinically actionable sex-specific therapeutic targets requires further research. Technical factors contributing to variability in differentially expressed gene profiles across studies also present a challenge, highlighting the need for continued investigation.

## 4. Efficacy and Potential Mechanisms of Exercise in Chronic Pain Relief

Effective chronic pain management is multifactorial and challenging. Drug therapy often falls short of expectations and can even cause severe side effects. This has led to an increase in research on non-pharmacological interventions therapies, among which exercise, defined as planned, structured, and repetitive bodily movement, is a highly recommended treatment for many chronic pain conditions [77]. Both human and animal studies support this idea. Human clinical trials provide essential evidence for the efficacy of exercise in reducing pain in individuals with different pain conditions (Table 1). However, they are ethically and technically limited in their ability to reveal the precise molecular and cellular mechanisms of these conditions. Therefore, preclinical animal models are invaluable for mechanistic investigation. These models are designed to recapitulate distinct pain phenotypes, such as inflammatory pain (via injection of Complete Freund’s Adjuvant (CFA)), neuropathic pain (via chronic constriction injury (CCI) of the sciatic nerve), or postoperative pain (via plantar incision). In these models, pain-like behaviors are measured as proxies for subjective pain, including stimulus-evoked hypersensitivity and spontaneous pain. This section will integrate findings from both human and animal research, using clinical studies to establish the therapeutic effects of different exercise types and preclinical studies to explore the underlying biological pathways. It is important to note that while animal models are crucial for discovering the mechanisms underlying chronic pain, the translation of these findings to humans requires careful consideration and further research.

### 4.1. Different Exercise Types in Chronic Pain Management

While various exercise types have demonstrated efficacy in chronic pain management, their underlying mechanisms can differ (Table 2). Aerobic exercise, such as walking, cycling, and swimming, remains one of the most extensively studied and widely prescribed interventions. The analgesic effect of aerobic exercise was hypothesized to be mediated by the endogenous opioid system, a premise supported primarily by findings from animal studies. However, a study by Bruehl et al. provided direct human evidence of this effect [88]. In this study, patients with chronic low back pain (CLBP) reported less pain after a six-week aerobic exercise intervention, and this benefit was no longer experienced when patients received naloxone, an opioid antagonist, proving that the analgesic effect was opioid-dependent. Still, other studies involving healthy human participants have observed elevated circulating beta-endorphin levels following aerobic and resistance exercise with blood flow restriction [89,90]. In addition, aerobic exercise may relieve chronic pain partly by improving depression and anxiety, which are common and clinically important moderators of pain outcomes. It has been demonstrated that 12 weeks of high-intensity aerobic exercise in CLBP reduced both pain intensity and psychological distress, indicating a mood-linked pathway to analgesia [91].

For conditions involving joint instability, such as knee osteoarthritis, resistance training appears to work through local mechanical effects [85]. By strengthening the muscles surrounding a joint (such as quadriceps and glutes), resistance training improves dynamic stability and reduces the load on the joint itself, thereby alleviating pain and improving joint function. During and after exercise, skeletal muscle functions as an endocrine organ that releases myokines into the circulation, thereby shaping the systemic immune and neurochemical milieu [92]. Among these, IL-6 rises acutely with resistance exercise in humans and, in an exercise context, exerts anti-inflammatory effects by inducing IL-10 and IL-1ra while suppressing pro-inflammatory cytokines such as TNF-α [93]. Repeated bouts of exercise appear to lower basal inflammation over time, indicating a shift toward an anti-inflammatory phenotype that could reduce the progression of nociceptive pain.

**Table 2 biology-14-01116-t002:** Comparison of exercise types for chronic pain management.

Exercise Type	Primary Goal	Potential Mechanisms of Action	Prescription Parameters	Considerations and Limitations	References
Aerobic Exercise	Improve cardiovascular endurance; reduce widespread pain; improve mood and sleep	Systemic anti-inflammatory effects (↓CRP, ↓TNF-α); release of endogenous opioids; improvement in mood (↑5-HT, ↑BDNF)	Intensity: moderate (able to talk but not sing) Frequency/Duration: 150 min per week can be divided into 30 min/day or 5 days/week	For severe arthritis or low physical capacity, start with low-intensity; “start low, go slow” principle is important	[88,94,95,96]
Resistance Training	Increase muscle strength and function; improve localized pain; improve metabolic level	Release of anti-inflammatory myokines; improvement in local biomechanics; reduction in joint load and increase in pain threshold	Intensity: Weight that can be lifted with 8–15 repetitions; Frequency: 2–3 times/week, primarily for targeted muscle groups	More technical requirements; incorrect posture may cause injury; professional guidance recommended; not suitable for acute inflammation	[95,97]
Mind–Body Exercise	Improve central sensitization; reduce anxiety, fear, and avoidance; enhance body awareness and balance	Regulation of ANS (↑parasympathetic activity); enhancement in descending inhibitory pathways (↑GABA, ↑5-HT); reduction in maladaptive pain-related cognition	Intensity: Low with focus on breathing and movement coordination; Frequency/Duration: 2–3 times/week, with each lasting session 45–60 min	Effects influenced by physical confidence and psychological factors; limited mechanistic studies; more high-quality evidence needed	[98,99]

CRP, C-Reactive Protein; TNF-α, tumor necrosis factor-α; 5-HT, serotonin; BDNF, brain-derived neurotrophic factor; ANS, autonomic nervous system; GABA, gamma-aminobutyric acid; ↑: Increase; ↓: Decrease.

Mind–body exercises, such as yoga, Tai Chi, and Pilates, have garnered significant attention for its integrative approach, combining physical postures, breathing techniques, and mental focus. A mediation analysis indicated that decreases in negative affect, fear of movement, and pain catastrophizing account for a meaningful portion of the analgesic effect after Pilates exercise [100]. The unique mechanisms underlying mind–body exercise are thought to directly modulate neurochemical systems integral to pain and mood. Evidence using magnetic resonance spectroscopy indicates that yoga practice can elevate thalamic concentrations of gamma-aminobutyric acid (GABA), the principal inhibitory neurotransmitter in the brain [98]. An enhancement in GABAergic tone may counteract the neuronal hyperexcitability that underpins the phenomenon of central sensitization. While there is less direct evidence for 5-HT modulation caused by mind–body exercise in chronic pain populations, it represents a plausible contributing mechanism given the well-documented improvements in comorbid anxiety and depression, and the established role of 5-HT in both mood regulation and descending pain modulatory pathways [99,101]. Another primary mechanism is the rebalancing of the autonomic nervous system (ANS). Chronic pain states are frequently associated with sympathetic hyperactivity, a form of dysregulation manifesting as a reduction in heart rate variability (HRV), reflecting impaired parasympathetic control over cardiac function [102,103]. Studies have demonstrated that both Tai Chi and Yoga can enhance parasympathetic tone, as evidenced by improvements in HRV parameters [104,105], which may contribute to their analgesic effects. Future investigations could benefit from incorporating ANS-related biomarkers to develop more individualized and effective exercise prescriptions for chronic pain management.

Despite the established benefits of “exercise as medicine”, the single greatest challenge preventing its standardized implementation of “exercise as medicine” is the lack of consensus on optimal dosing. The FITT principle (Frequency, Intensity, Time, Type) provides a framework for prescription, but the literature is characterized by significant heterogeneity [106]. Furthermore, due to pain phenotype differences, it is critical to establish stratified rehabilitation strategies that use objective biomarkers, such as inflammatory or neuroimaging signatures, to match patients to the most suitable interventions. In addition, most current research relies on subjective, self-reported outcomes. A stronger causal link needs to be established between these clinical improvements and the objective biomarker changes measured in laboratory settings.

### 4.2. Mechanistic Insights from Preclinical Animal Research

Preclinical animal models of chronic pain (e.g., nerve injury, inflammation) provide a platform to investigate the direct biological mechanisms of exercise-induced analgesia. These models allow for an invasive level of inquiry that is not possible in humans, enabling us to map the biological adaptations induced by exercise directly onto the key nodes of pain pathophysiology. As conceptually illustrated in Figure 3, exercise initiates a multi-system response that targets both the peripheral and central nervous systems. The following sections will detail these mechanisms, demonstrating how exercise actively suppresses neuroinflammation, enhances endogenous analgesia, and normalizes neuronal excitability, thereby providing a direct biological counter-response to the drivers of pain chronicity.

#### 4.2.1. Peripheral Mechanisms of Exercise-Induced Chronic Pain Relief

Systemic and Local Anti-Inflammation

A substantial body of evidence from animal models suggests that one of the most potent effects of exercise is the local and systemic modulation of the immune response at sites of injury. Macrophages are crucial local immune cells that polarize into distinct phenotypes based on their cytokine-secreting profiles: M1 macrophages (classically activated) release pro-inflammatory cytokines (IL-1β, IL-6, TNF-α), acting as key mediators of host defense while promoting hyperalgesia; conversely, M2 macrophages (alternatively activated) secrete anti-inflammatory cytokines (IL-10, IL-4, IL-1ra), facilitating tissue repair and exerting analgesic effects [107]. In rats, it has been demonstrated that prior voluntary wheel running exercise can both prevent the development of and reverse chronic pain following sciatic nerve injury [108]. In correlation with this, exercise suppressed the injury-induced increase in M1 macrophages within the sciatic nerve while concurrently promoting the expression of M2 macrophages. Further supporting this, Bobinski et al. observed that two weeks of treadmill exercise significantly reduced the proportion of M1 macrophages and markedly increased the M2 macrophage population in mice with sciatic nerve injury [109]. This shift was accompanied by significantly elevated local concentrations of the anti-inflammatory cytokines IL-4 and IL-1ra. Critically, IL-4 knockout mice failed to achieve exercise-induced pain relief, suggesting that IL-4 signaling is a critical component of the analgesic effects of exercise mediated by macrophage polarization. Similarly, studies in models of chronic muscle pain models have shown that pharmacological blockade of IL-10 receptors negated the analgesic benefits of exercise, whereas administration of IL-10 reduced pain behaviors [110]. Recent findings in a chronic muscle pain model further reveal that regular swimming prevented persistent hyperalgesia, which was associated with enhanced M2 macrophage polarization and increased expression of anti-inflammatory cytokines IL-4 and IL-10 [111].

Another important peripheral mechanism involves myokines, which are signaling molecules released by contracting skeletal muscle during exercise [112]. An emerging exercise-induced myokine with potential in chronic pain management is irisin, which has shown efficacy in reducing pro-inflammatory cytokines and increasing anti-inflammatory cytokines in preclinical rodent studies [113]. Xie et al. found that irisin treatment alleviated chronic constriction-injury-induced mechanical allodynia and thermal hyperalgesia; meanwhile, the levels of pro-inflammatory cytokines (IL-1β, IL-6 and TNF-α) mediated by NF-κB in the spinal cord were decreased [114]. Rahman et al. also reported inflammatory-pain-relieving behavior of irisin in mice [115]. The inflammatory condition of the DRG was improved as irisin upregulated the expression of M2 macrophage markers (IL-4 and IL-10) and downregulated that of M1 macrophage markers (IL-1β, IL-6 and TNF-α). These findings suggest that active muscle tissue functions as an endocrine organ, releasing anti-inflammatory signals that contribute to pain relief.

Considering the existing literature on this topic, it can be proposed that exercise can promote a shift in macrophage polarization away from an M1 state towards an M2 state, leading to reduced production of pro-inflammatory mediators and increased release of anti-inflammatory cytokines at peripheral injury sites or in a systemic environment to promote analgesia.

2.Pain-Associated Ion Channels

The interplay between immune responses and the aberrant activation of specific ion channels in primary sensory neurons is critical for peripheral sensitization. In addition to improving inflammatory conditions, regular exercise may alter the expression levels of key pain-associated ion channels and reconfigure their function in DRG neurons. In animal studies, it has been reported that regular exercise alleviated mechanical allodynia caused by a high-fat diet, possibly by decreasing TRPA1 mRNA expression and the level of AKAP150 (a protein involved in TRPV1 membrane translocation and insertion), and also caused a reduction in inflammatory mRNAs [116]. Similarly, a decrease in the protein expression of TRPV1 and TRPM8 ion channels in the DRG was detected in diabetic mice subjected to exercise compared with a diabetic sedentary group [117]. These results point to a possible mechanism whereby exercise disrupts the bidirectional vicious cycle of inflammation and the hyperactivation of ion channels, particularly TRP channels, thereby relieving chronic pain.

#### 4.2.2. Central Mechanisms of Exercise-Induced Chronic Pain Relief

The Endogenous Opioids System

Since the introduction of the idea of central sensitization, a large body of research has emphasized that central mechanisms and central adaptations are critical mediators in the field of exercise-induced analgesics. One of the most widely studied central mechanisms of exercise-induced analgesia is the activation of the endogenous opioids system, which refers to the naturally occurring opioid peptides (endorphins, enkephalins and dynorphins) produced primarily within CNS, along with the specific opioid receptors (mu/delta/kappa-opioid receptors) they bind to, and the physiological pathways they regulate. It essentially acts as the body’s own internal system for managing pain, stress, and reward. Animal studies provide robust and consistent evidence for this mechanism. In rats with neuropathic pain, increased met-enkephalin and leu-enkephalin were found in PAG and RVM, two critical nodes in opioid-induced analgesia, following a single bout of exercise as well as exercise training [118,119]. Moreover, in a mouse model of chronic muscle pain, five days of voluntary wheel running activity effectively prevented the development of hyperalgesia in wild-type mice, but this protective effect was absent in mice lacking the mu-opioid receptor, indicating the observed exercise-induced analgesia under these conditions is dependent upon the activation of mu-opioid receptors [120]. While opioid receptors are distributed in the peripheral tissue, the CNS seems to be more responsible for exercise-induced chronic pain relief [119]. Specifically, Brito et al. demonstrated that wheel running for eight weeks blocked the production of hyperalgesia in a mouse model of chronic muscle pain. In addition, systemic naloxone, intra-PAG naloxone, and intra-RVM naloxone all reversed the analgesic effects of regular exercise, whereas systemic injection of the peripherally restricted opioid antagonist naloxone methiodide had no impact, suggesting effects of exercise at the central sites, but not the peripheral site, on mediated the endogenous opioid system [121].

Thus, exercise-induced released as a result of exercise, notably beta-endorphins, diffuse to bind with opioid receptors, primarily mu-receptors, situated on neurons in descending pain modulatory pathways such as the spinal dorsal horn, PAG, and RVM, thereby reducing the sensation of pain. However, most studies focus on the immediate effects of acute exercise on endogenous opioid system activation, and the effects of long-term exercise interventions are not completely understood.

2.The serotonergic system

Alongside the opioid system, exercise is known to modulate the activity of the serotonergic system to achieve analgesic effects, where increased release of 5-HT release seems to play a dominant role. Animal studies consistently show that exercise enhances central serotonergic activity. Cumulative evidence indicates that both acute and chronic exercise can enhance central serotonergic activity, often reflected by increased synthesis, release, and turnover of 5-HT (indicated by elevated 5-HIAA levels) in regions such as the raphe nuclei and the spinal dorsal horn [122,123,124]. Recent findings show that regular aerobic exercise can induce pain relief in chronic inflammation, while elevated levels of 5-HT and modulated synaptic plasticity were found to occur in the ACC, a key area of the brain for pain information processing [125]. An analgesic contribution of 5-HT7 receptors has been suggested, with studies indicating that their blockade or knockdown partially suppressed the beneficial effects of exercise on pain at different time points. A plausible mechanism is that exercise boosts 5-HT release, which then acts on 5-HT7 receptors to potentially modulate the ACC–spinal cord circuitry, leading to inhibited spinal dorsal horn neuron activity in response to pain and reduced afferent signaling, resulting in analgesia. However, it must be emphasized that this proposed pathway remains speculative at present. Recent work also points towards a significant functional interplay between the endogenous opioids system and the 5-HT system. Pharmacological studies have shown that 5-HT receptor agonists can enhance the antinociceptive effects of opioids [126]. Furthermore, signals from both systems often converge onto the same populations of spinal neurons. For instance, within the RVM, morphine administration from PAG induces opioid receptor activation and modulates the activity of descending 5-HT neurons, potentially increasing their inhibitory nociceptive output to the spinal cord [127,128]. Of note, blockade of 5-HT3 receptors in the spinal cord reduced the analgesic effects of morphine in normal rats but facilitated morphine-induced analgesia in a model of rat neuropathic pain, which may account for functional changes in spinal GABA_A_ receptors [129]. Researchers also examined 5-HT transporter expression in the RVM, comparing mu-opioid receptor knockout mice with wild-type mice [120]. The results showed that five days of voluntary wheel running prevented the increase in 5-HT transporter arising from chronic muscle pain in wild-type mice; however, alterations in the 5-HT transporter caused by wheel running did not occur in mu-opioid receptor knockout mice, suggesting that exercise activates mu-opioid receptors, which in turn leads to a reduction in 5-HT transporter expression in the RVM. Similarly, in a chronic muscle pain model, wheel running activated opioid receptors and the 5-HT system in the PAG and RVM, reducing muscle hyperalgesia, with reduced 5-HT transporter expression in animals with muscle insult [121]. Studies collectively indicate that the endogenous opioid system and 5-HT systems cooperate to mediate exercise-induced analgesia and chronic pain relief.

3.The NMDA receptor

The NMDA receptor is a crucial excitatory glutamate receptor, and its hyperactivity is widely recognized as a critical contributor to central sensitization, a key mechanism underlying the maintenance of chronic pain states. In particular, NMDA receptors activation is essential for the LTP observed in pain pathways, which underpins allodynia (pain from non-painful stimuli) and hyperalgesia (increased pain sensitivity) in chronic pain states [130]. Exercise may help modulate the function and expression of NMDA receptors, thereby contributing significantly to a reduction in excitatory transmission [45]. In animal studies, research has demonstrated that in a model of exercise-induced pain using by pH 5.0 injections, blocking NMDA receptors in the medullary raphe nuclei (such as nucleus raphe obscurus and nucleus raphe pallidus) during exercise prevented the development of hyperalgesia, suggesting that NMDA receptor activation in these brain regions is essential for enabling the interaction between exercise and pain [131]. Further, exercise might change the phosphorylation state or expression levels of key NMDA receptor subunits, particularly the NR1 subunit, which is strongly implicated in pain plasticity [132]. Such changes could lead to decreased channel conductance or accelerated internalization of the receptor, effectively reducing neuronal excitability [133,134]. This idea is supported by the findings that wheel running blocks muscle-injury-induced increases in NR1 phosphorylation in the RVM [135]. Additionally, four weeks of treadmill exercise alleviated persistent postsurgical pain caused by skin/muscle incision and retraction surgery compared with postsurgical sedentary rats, with a decrease in NR1 expression in the spinal cord involved in this process [136].

4.Glial cells

As previously discussed, chronic pain is frequently associated with the persistent activation of glial cells, primarily microglia and astrocytes, within key pain processing regions of the CNS, especially the spinal cord dorsal horn. In response to injury or persistent nociceptive input, these glial cells transition to a reactive state, releasing a cascade of pro-inflammatory mediators, including IL-1β, TNF-α, and IL-6, as well as chemokines and other signaling molecules. This neuroinflammatory milieu directly enhances neuronal excitability, facilitates synaptic transmission in pain pathways, and impairs inhibitory controls, thereby contributing significantly to the development and maintenance of central sensitization. Emerging evidence in animals strongly suggests that regular physical activity can effectively counteract astrocytes and microglial hyperactivity in the dorsal horn, leading to long-lasting analgesia [137,138,139,140]. Recent studies have endeavored to elucidate the exact molecular mechanism behind neuroimmune function [141]. Specifically, in neuropathic pain models, increased expression of astrocyte markers and enhanced complexity/length of astrocytic processes in the spinal dorsal horn indicated reactive astrogliosis, while four weeks of low-intensity treadmill exercise significantly attenuated this astrocyte reactivity and suppressed the upregulation of pro-inflammatory cytokines (IL-1β and TNF-α) [142]. Meanwhile, complement component 3 (C3) knockout mice and exogenous C3 administration confirmed that exercise-induced analgesia is directly correlated with downregulation of C3 in reactive astrocytes, suggesting that exercise relieved neuropathic pain by suppressing C3 expression in reactive spinal astrocytes, thereby reducing neuroinflammation and astrocyte hyperactivity. The role of microglia has also been demonstrated recently, with a study indicating that high-intensity swimming alleviates chronic post-ischemia pain by activating the spinal pro-resolving lipid mediator axis, which inhibits microglial activation and shifts the cytokine balance from pro-inflammatory (decreasing IL-1β, IL-6, and TNF-α) towards anti-inflammatory (increasing IL-4 and IL-10); this could be relevant for managing complex regional pain syndrome type I [143]. Again, while promising, these results should be interpreted with caution as they stem solely from studies in mice. For example, a review on rheumatoid arthritis concluded that clinical inflammatory markers and inflammatory cytokines did not differ between people with and without rheumatoid arthritis in the context of exercise [144].

## 5. Conclusions

Chronic pain presents significant clinical challenges due to its multifaceted classifications and complex dysregulations of multiple physiological systems. The evidence reviewed in this paper demonstrates that exercise is not merely a relieving therapy but also a targeted, comprehensive biological intervention. This review has systematically shown that the mechanisms of exercise-induced analgesia function as a direct counter-response to the core pathologies of chronic pain, where pain chronicity is caused by peripheral sensitization driven by local inflammation and central sensitization characterized by glial activation; aberrant synaptic plasticity; and dysregulation of descending modulatory pathways. Evidence from animal studies has shown that the therapeutic power of exercise lies in the following: Peripherally, exercise fosters a systemic anti-inflammatory environment, notably by promoting the polarization of macrophages and activating the release of myokines from skeletal muscle. Centrally, exercise activates the endogenous opioid and serotonergic systems, thereby enhancing descending pain inhibition. Additionally, it appears to normalize neuroplasticity by modulating the function of NMDA receptors and prevent the development of neuroinflammation by suppressing the reactivity of microglia and astrocytes. This integrated effect provides a solid biological rationale for its clinical application.

## 6. Future Directions

Translating our findings into clinical practice presents several key challenges. One significant obstacle is the limitations of preclinical animal models. The majority of animal studies rely on genetically homogeneous, young male rodents [145], a model that does not reflect the diversity of factors that influence chronic pain in humans, such as age, sex, genetics, and comorbidities. This is a crucial distinction, as the neuro-immune interactions involved in pain are known to differ between males and females, with distinct mechanisms operating in each sex [64]. These differences necessitate sex-specific therapeutic strategies in the future. Furthermore, animal models cannot replicate the psychological and cognitive aspects of chronic pain, such as kinesiophobia, pain catastrophizing, and placebo effects, which profoundly influence rehabilitation outcomes [146]. Therefore, future research must incorporate more sophisticated models, including animal models with greater relevance to humans and reverse-translation approaches, in which clinical insights inform laboratory experiments. These strategies will ensure that the mechanisms identified are both relevant and translatable to human conditions.

The current understanding of exercise as an intervention for chronic pain highlights the necessity for a tailored approach rather than a universal prescription approach [77]. Given the diversity in pain phenotypes, there is an urgent need for studies that stratify patients based on their primary pain mechanism, such as nociceptive, neuropathic, or nociplastic pain, and examine how different exercise modalities affect each phenotype. In addition, personalized exercise therapy requires objective stratification and prediction [147]. Current evidence lacks reliable tools to anticipate individual responses to specific programs, making it necessary to identify and validate biomarkers that link mechanisms to outcomes. Future studies should aim to identify inflammatory markers, such as cytokine levels, genetic factors, or functional neuroimaging signatures, that can be used to tailor exercise interventions. Finally, there is no definitive optimal dose of exercise that achieves analgesia without provoking pain flares [148]. Therefore, rigorous dose–response studies are needed to establish therapeutic windows for various exercise types and pain conditions.

## Figures and Tables

**Figure 1 biology-14-01116-f001:**
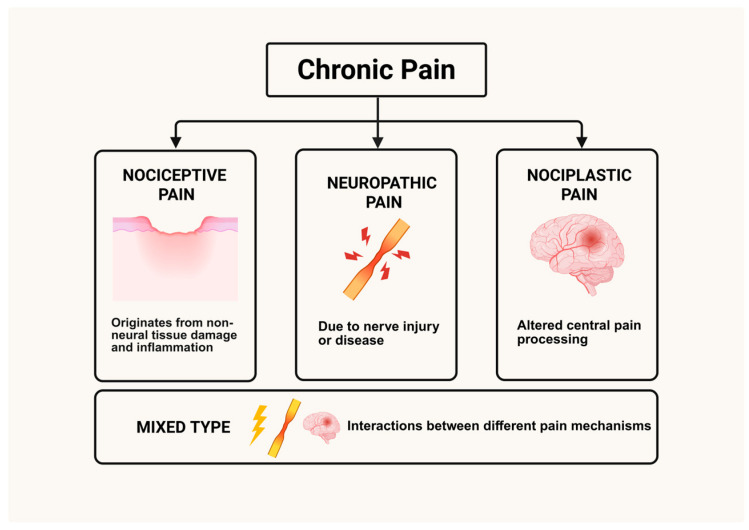
Classification of chronic pain.

**Figure 2 biology-14-01116-f002:**
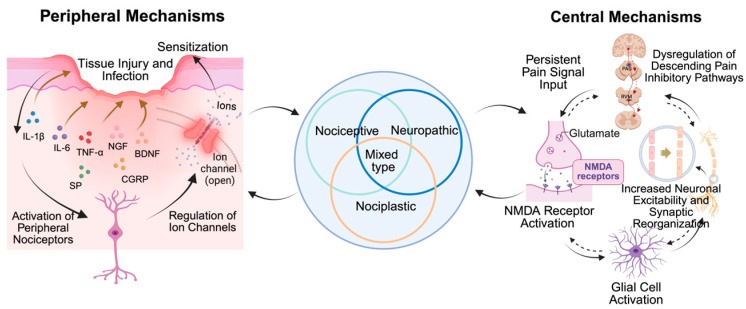
Molecular mechanisms of chronic pain. Peripheral ion channel regulation, pro-inflammatory mediator release, central neural plasticity changes, and glial cell activation all contribute to the establishment of chronic pain state. IL-6, interleukin-6; IL-1β, interleukin-1β; TNF-α, tumor necrosis factor-α; NGF, nerve growth factor; BDNF, brain-derived neurotrophic factor; NMDA, N-methyl-D-aspartate; SP, substance P; CGRP, calcitonin gene-related peptide.

**Figure 3 biology-14-01116-f003:**
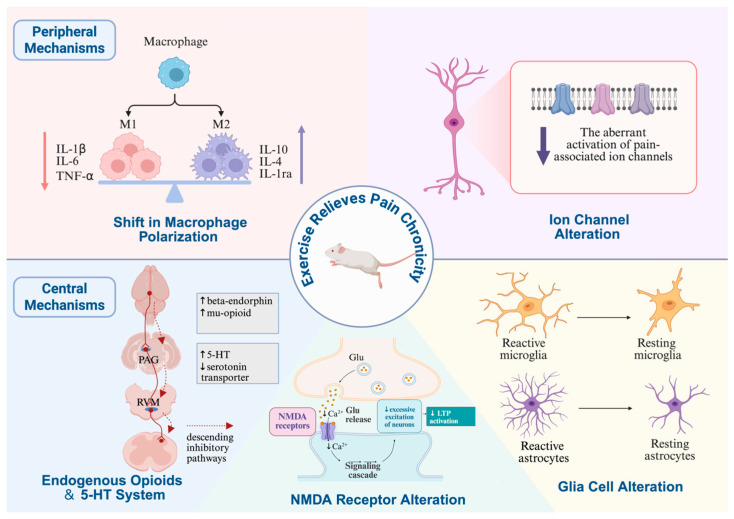
An overview of the central and peripheral mechanisms underlying exercise-induced analgesia from animal studies. Exercise counteracts chronic pain pathology by activating descending inhibitory pathways, suppressing central neuroinflammation, and promoting a peripheral anti-inflammatory environment. IL-1β, interleukin-1β; IL-6, interleukin-6; TNF-α, tumor necrosis factor-α; IL-10, interleukin-10; IL-4, interleukin-4; IL-1ra, interleukin-1 receptor antagonist; 5-HT, serotonin; PAG, periaqueductal gray; RVM, rostral ventromedial medulla; NMDA, N-methyl-D-aspartate; Glu, glutamate; ↑: Increase; ↓: Decrease.

**Table 1 biology-14-01116-t001:** Summary of studies investigating exercise-induced analgesia in individuals with different pain conditions.

Reference	Pain Condition	Sample (N)	Exercise Type	Exercise Form	Intensity	Duration	Pain Outcome and Findings
Tilbrook et al. (2011) [78]	CLBP	313	Mind–body	Yoga	Gradually progressing	1 × wk, 12 wks	No change in pain intensity
Bruehl et al. (2021) [79]	CLBP	83	Aerobic	Not specified	70–85% HRR	3 × wk, 6 wks	↓Pain intensity; ↓HPI; ↑HPT; no change in PPI
Saleem et al. (2025) [80]	CLBP	140	Mind–body	Yoga	Moderate	5 × wk, 12 wks	↓Pain intensity
Hooten et al. (2012) [81]	FM	72	Aerobic vs. Resistance	Cycling vs. Not specified	Progressive, to maximal tolerance	7 × wk, 3 wks	Both groups showed equivalent ↓Pain severity
Bjersing et al. (2012) [82]	FM	49	Aerobic	Walking	Moderate to high vs. low	2 × wk, 15 wks	No change in pain intensity; ↑PPT
Andrade et al. (2019) [83]	FM	54	Aerobic	Aquatic training	80–110% VAT HR	2 × wk, 16 wks	↓Pain intensity; ↑PPT
Wang et al. (2016) [84]	KOA	204	Mind–body	Tai Chi	—	2 × wk, 12 wks	↓Pain intensity
Aguiar et al. (2016) [85]	KOA	22	Resistance	—	80% 10 RM	12 weeks	↓Pain intensity
De Araujo Cazotti et al. (2018) [86]	chronic neck pain	64	Mind–body	Pilates	6–12 repetitions	2 × wk, 12 wks	↓Pain intensity
Stegner et al. (2021) [87]	CMP	54	Resistance	Machines	Low-start, gradual progression	2 × wk, 16 wk	No change in pain intensity

CLBP, chronic low back pain; FM, fibromyalgia; KOA, knee osteoarthritis; CMP, chronic widespread musculoskeletal pain; ×wk, times per week; wks, weeks; HPI, heat pain intensity; HPT, heat pain threshold; PPI, pressure pain intensity; PPT, pressure pain threshold; HRR, heart rate reserve; VAT, ventilatory anaerobic threshold; RM, repetition maximum; ↑: Increase; ↓: Decrease.

## Data Availability

No new data were created or analyzed in this study. Data sharing is not applicable.

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
