# Peer review of "Pain Chronicity and Relief: From Molecular Basis to Exercise-Based Rehabilitation"

_biology, 2025, doi:10.3390/biology14091116_

Round 1

Reviewer 1 Report

Comments and Suggestions for Authors

‘’Nociceptive pain originates from direct non-neural tissue tissue damage and inflammation’’

Comment 1: One of the ''tissue'' should be deleted (line 112)

‘’…the central nervous system seems more responsible for exercise-induced chronic pain relief [95].’’

Comment 2: CNS should be written instead of the central nervous system (line 360)

‘’…like ACC and amygdala for neuropathic pain [70–72].’’

Comment 3: The open form of ACC should be written (line 248)

‘’… found in anterior cingulate cortex (ACC), a key brain area for pain information processing [103].’’

Comment 4: Just ACC is enough (line 383)

‘’Recent studies endeavor to seek the the exact molecular mechanism behind neuroimmune function [121].

Comment 5: One of the ''the'' should be deleted (line 452)

Comment 6: The article mainly focuses on the biochemical aspects of the connection between exercise and chronic pain. The conclusion part states, "This review explored the fundamental pathogenic mechanisms underlying pain chronicity across the central and peripheral nervous systems." However, the article does not mention the autonomic nervous system and its dysfunction anywhere. Autonomic nervous system  (ANS) dysfunction plays a crucial role in the pathophysiology of chronic pain, and exercise can regulate ANS dysfunction. In fact, patients with various chronic pain disorders can be prescribed optimal and specific exercises based on their ANS activity level. Adding ANS dysfunction in detail may divert the article from its purpose, but authors should at least touch on this topic, even if only in a few sentences.

Author Response

Reviewer#1.

Line 112 - One of the “tissue” should be deleted

⇒ Thanks for your comments. We have corrected the typo by removing the redundant word “tissue”.

Line 360 - CNS should be written instead of “the central nervous system”

⇒ Thanks for your comments. We have replaced “the central nervous system” with “CNS” to improve clarity and conciseness.

Line248 and Line 383 - The open form of ACC

⇒ Thanks for your comments. We have revised the text to read “anterior cingulate cortex (ACC)” upon its first mention for clarity.

Line 452 – One of the “the” should be deleted

⇒ Thanks for your comments. We have removed the duplicate “the” to correct the sentence.

The role of ANS

⇒ Thanks for your comments. We acknowledge that autonomic nervous system (ANS) dysfunction plays a significant role in chronic pain regulation and that exercise can modulate ANS activity. We have added a brief mention of this topic in the section “4.1 Diverse Exercise Types in Chronic Pain Management” to acknowledge its relevance and suggest it as an important direction for future research.

Reviewer 2 Report

Comments and Suggestions for Authors

Comment to the Author: I commend the authors for putting this comprehensive paper together. This is an interesting study and will be a valuable contribution to the field, but needs a re-think prior to being ready for publication on its intended purpose. The suggested improvements listed as comments below are intended to strengthen the paper further. On a general note, this paper needs a grammatical check (particularly in the introduction), as several areas lose their effectiveness and/or meaning due to grammatical errors. It also needs to be clearly explained where this paper sits in the field. There are existing publications on mechanisms underlying pain pathways (including from the molecular level). Some of these publications have not been cited in this paper, which is concerning. Please outline the processes undertaken for the inclusion/scope of studies used in this review, and consider revising. What makes this paper novel and an important addition to the field? What publications already cover the molecular basis for pain and what makes this paper different? One of the key highlights of this paper is the inclusion of exercise-based responses on chronic pain pathways, so the novelty of this aspect should be highlighted. However, exercise prescription as a therapeutic pathway did not feature enough in the article. For example, some aspects mentioned in the conclusion received little attention throughout the paper, such as exercise modality and specifics of training programs. Since exercise has been long-studied in the context of chronic pain, this needs to be highlighted more in the paper. This notion contributes to the confusion around where this paper will sit in the field. In addition, there is an interchange in the latter half of the paper between animal studies and humans. It is difficult to discern in many places whether the content is talking about human or animal biology. Thus, the generalisability of the pathophysiology to humans is lacking in places, which undermines the quality of the paper.

Section 2:

  • Why is there no mention of IL-6 in Fig 2 and the corresponding paragraph?
  • What about idiopathic pain?
  • What about somatic, or visceral, pain?
  • Please explain how the persistence of stimuli causes nerve damage

Section 3:

  • Exercise is not always prescribed and/or supervised but can still constitute exercise. Please revise the definition of what constitutes ‘exercise’.
  • Fig 3 shows a person but much of the content alludes to rodent models – this mismatch needs to be considered and addressed throughout the paper. Fig 3 also has a typo in glia cells, and is missing several abbreviations in the legend.
  • What is the relevance of animal models here? Is it due to lack of studies in pain pathology in humans, or are there studies that can be used from humans instead? Is the whole section about animal studies? Be explicit when swapping from humans to animals and vice versa so it is clear to the reader. There needs to be greater distinguishing between them, with a clear rationale for the context. There is also no mention of the generalisability of pain pathology mechanisms between humans and various animal models.
  • Pain is a subjective experience, so how was this measured in an animal model and thus how can this be applied to humans?
  • Is any of this section about people?
  • It is extremely disappointing that most of the section pertaining to exercise rehabilitation is also talking about exercising rodents. There is a wealth of research of exercise-based rehabilitation for chronic pain conditions in people that should be discussed here (it warrants an entire section alone). Otherwise, the focus on animal models needs to be stated in the title and throughout the paper.

Section 4:

  • Again, lack of specificity between people and animals.
  • Exercise programs in rehabilitation settings haven’t been discussed at all (i.e. exercise prescription). There is a mention of a significant gap in the research for specific exercise prescription for pains with chronic pain conditions, but how do you know this? The paper has barely reviewed this, so such a statement is unwarranted. There is actually a gap in this area of the research, but it is much more specific than what is stated here.
  • Much of the conclusion is actually missing from the content of the body of the paper.
  • There have been many studies on anxiety and depression and the benefits of exercise as a therapeutic tool.
  • Very little mention of people here either.
Comments on the Quality of English Language

As above

Author Response

Language and grammar issues reduce clarity.

⇒ Thanks for your comments. We have carefully revised the manuscript for grammar, syntax, and clarity. This included a comprehensive grammatical check performed by both the authors and an external professional language-editing service. The revised version has also been submitted to MDPI’s professional English editing service (English-99155)

Paper’s position in the field is unclear; key literature missing.

⇒ Thanks for your comments. We have now revised the introduction to clearly position this paper within the existing body of literature. Additional citations to key publications covering molecular mechanisms of pain have been included, and we have explicitly clarified how our review complements and extends previous works, with a particular emphasis on the integration of exercise-based responses into chronic pain pathways.

The processes for inclusion/scope of studies in the review were not outlined.

⇒ Thanks for your comments. We have now added a dedicated Section 2. Search Strategy describing the inclusion and exclusion criteria, databases searched, search terms used, and the selection process for studies considered in this review. This ensures transparency and reproducibility of our review process.

The novelty of the paper is not clear, especially compared to existing literature on the molecular basis of pain.

⇒ Thanks for your comments. We have revised the abstract and introduction to clearly highlight the novelty of our work, which lies in synthesizing molecular mechanisms of chronic pain with evidence on exercise-induced modulation of these pathways. Unlike existing reviews, our work provides a cross-disciplinary integration of pain biology and exercise prescription, offering a unique perspective for translational research and clinical application.

The exercise-based responses on chronic pain pathways, which are a key highlight, were not sufficiently developed.

⇒ Thanks for your comments. In the revision, we substantially expanded and reorganized the exercise-focused mechanistic content to provide a pathway-level account of how exercise modulates chronic pain:

  1. New integrative section on exercise efficacy and mechanisms - We refined Section 4, “Efficacy of Exercise for Chronic Pain Relief and its Potential Mechanisms,” to elucidate clinical outcomes with mechanistic pathways.
  2. Modalities and dosing made explicit - We introduced Section 4.1, “Different Exercise Types in Chronic Pain Management,” detailing aerobic, resistance and mind-body, with typical intensity/frequency parameters. We created Table 2, “Comparison of Exercise Types for Chronic Pain Management,” summarizing putative targets and physiological systems involved.
  3. Human evidence summarized across conditions - We compiled clinical studies into Table 1, “Summary of studies investigating exercise-induced analgesia in different pain conditions,” covering CLBP, fibromyalgia, KOA, etc.
  4. Mechanistic depth in peripheral and central pathways - We refined Section 4.2, “Mechanistic Insights from Preclinical Animal Research,” and subdivided it into:

4.2.1 Peripheral mechanisms with expanded subsections on “Systemic and Local Anti-Inflammation” (emphasizing cytokine/immune milieu at injury sites as well as systemic effects) and “Pain-Associated Ion Channels” (e.g., modulation of TRP channels).

4.2.2 Central mechanisms detailing the endogenous opioid system, serotonergic system, NMDA receptor-dependent plasticity, and glial cell modulation, explaining how exercise enhances descending inhibition, adjusts neurotransmission, and attenuates neuroinflammation.

Together, these additions provide a much more developed, modality-specific, and mechanistically grounded treatment of how exercise influences chronic pain pathways, directly addressing the reviewer’s concern

The conclusion mentions aspects such as exercise modality and specifics of training programs, but these were not addressed adequately in the main text.

⇒ Thanks for your comments. We introduced Section 4.1, “Different Exercise Types in Chronic Pain Management,” detailing aerobic, resistance and mind-body, with typical intensity/frequency parameters. We created Table 2, “Comparison of Exercise Types for Chronic Pain Management,” summarizing putative targets and physiological systems involved.

The distinction between human and animal studies is unclear.

⇒ Thanks for your comments. In the revised manuscript, we have carefully clarified the distinction between human and animal studies throughout, to make the scope and generalizability of findings more transparent. We restructured Section 4 so that human studies and animal studies are presented in distinct paragraphs, with transitional sentences making it explicit when the discussion shifts from clinical to preclinical evidence. In the introduction to Section 4, we added a sentence emphasizing that while animal models are invaluable for mechanistic insights, translation to human populations requires caution, and results may not be directly generalizable without further validation. We summarize human clinical studies in Table 1 (“Summary of studies investigating exercise-induced analgesia in individuals with different pain conditions”), while mechanistic findings from animal studies are described separately in Section 4.2 (“Mechanistic Insights from Preclinical Animal Research”).

Section 2

About idiopathic pain, somatic, and visceral pain

⇒ Thanks for your comments. We recognize that although our classification is primarily based on mechanism, clarifying its relationship with the source-based clinical classifications of somatic and visceral pain improves the manuscript’s accessibility. Therefore, in Section 3.1, we have added a sentence following the definition of “Nociceptive pain” to explain that it is clinically subdivided into somatic and visceral pain (Line 131). Similarly, we have now clarified the important link between the concept of “idiopathic pain” and “nociplastic pain”. After defining “nociplastic pain” in Section 3.1, we have added a sentence to explain that this new classification provides a neurobiological basis for many chronic pain syndromes (e.g., fibromyalgia) that were previously considered “idiopathic” (Line 147).

Missing mention of il-6 in figure 2 and corresponding paragraph

Thanks for your comments. IL-6 is a critical pro-inflammatory cytokine involved in the inflammatory response in chronic pain conditions, and its omission in Figure 2 is indeed an oversight. We have updated both Figure 2 and the corresponding paragraph to include IL-6 as a key mediator.

Explanation of how the persistence of stimuli causes nerve damage

Thanks for your comments. In response to this, in Sections 3.2 and 3.3, we have included a more detailed discussion on the roles of axonal reflex (AR) and dorsal root reflex (DRR) in the development and maintenance of chronic pain (Line 199 and Line 245). These reflexes represent critical mechanisms that mediate peripheral pain responses and contribute to the propagation of pain signals from the periphery to the central nervous system. The axonal reflex refers to the activation of nociceptors in the periphery, which leads to neurogenic inflammation and pain amplification. The dorsal root reflex involves the re-excitation of nociceptive fibers at the dorsal root, contributing to the persistence of pain and central sensitization.

Section 3

Unclear definition of “exercise”

⇒ Thanks for your comments. In response, we have clarified the definition of “exercise” in the revised manuscript seen Line 305. We emphasize that exercise is not limited to prescribed or supervised activities but also includes general physical activity and informal exercise routines. This broadens the scope to include activities like walking, yoga, and non-supervised exercise, as long as they aim to improve physical health.

On Figure 3 inconsistencies and typos

⇒ Thanks for your comments. We have addressed the mismatch between the human subject in Figure 3 and the content discussing animal models. Figure 3 has been updated to clearly distinguish between the depiction of human studies and animal model research. And the typo in “glia cells” has been corrected. Additionally, we have added missing abbreviations in the figure legend for clarity.

On the relevance of animal models

⇒ Thanks for your comments. In the revised manuscript, we have made the following adjustments to clarify the use of animal models:

  1. Justification for Animal Models: We explicitly discuss why animal models are utilized in pain research. While human studies are indeed essential, animal models allow for in-depth exploration of biological and molecular mechanisms that are difficult to assess directly in humans due to ethical or technical limitations. In Section 4.2, we separate animal studies from human research and discuss their distinct contributions.
  2. Generalizability: We address the generalizability of pain pathology mechanisms between humans and animals in the revised manuscript. We acknowledge that while animal models provide valuable insights into pain mechanisms, the results must be carefully interpreted when applying them to human conditions in Section 5.

On pain subjectivity and measurement in animal models

⇒ Thanks for your comments. We recognize that pain is a subjective experience and have clarified how it is measured in animal models. In the beginning of Section 4, we explain that pain in animal models is typically measured through stimulus-evoked hypersensitivity and spontaneous pain. We emphasize the validity of these measurements as indirect indicators of pain experience.

Section 4

Lack of specificity between people and animals

⇒ Thanks for your comments. We revised Section 5 Conclusions and Future Directions to clearly separate preclinical mechanisms from human implications. We added explicit statements on generalizability limits, including sex differences and psychosocial modifiers.

Exercise programs in rehabilitation settings (exercise prescription) are not discussed

⇒ Thanks for your comments. Section 5 now includes rehabilitation-oriented content: phenotype-informed choice, optimal dose, and monitoring for pain flares. This adds the missing bridge from mechanisms to clinical implementation.

The claim of a “significant gap” in specific exercise prescriptions is unwarranted

⇒ Thanks for your comments. We removed the broad claim and replaced it with focused, evidence-anchored priorities that are supported by the material reviewed: phenotype-stratified trials (nociceptive, neuropathic, nociplastic), biomarker-guided prediction to identify likely responders, and rigorous dose–response work to delineate therapeutic windows by condition. The revised text now reflects what the paper actually reviewed.

Many studies show exercise benefits for anxiety and depression, but this is not reflected

⇒ Thanks for your comments. We have clarified that chronic pain frequently co-occurs with anxiety and depression and we have expanded on the mechanisms of various exercises, highlighting that their analgesic effects are significantly mediated by psychological factors in Section 4.1.

Reviewer 3 Report

Comments and Suggestions for Authors

The study presents an integrative review that lists some pain pathways and how physical exercise can help control chronic pain. I missed a subsection on the dorsal root reflex and axonal reflex, as well as physical exercise on these reflexes.

Author Response

A subsection on the dorsal root reflex and axonal reflex, as well as physical exercise on these reflexes

⇒ Thanks for your comments. We have addressed it by adding concise subsections on both reflexes in the mechanisms part of the manuscript: the axonal reflex (AR) is now described under Peripheral Mechanisms (Section 3.2), and the dorsal root reflex (DRR) is presented under Central Mechanisms (Section 3.3). These additions clarify their definitions and roles in chronic pain pathophysiology.

We did not extend Section 4 (exercise mechanisms) to discuss direct effects of exercise on AR or DRR. This was a deliberate choice for two reasons. First, current evidence directly testing exercise-induced modulation of AR or DRR is limited, with most data being indirect through upstream or downstream processes such as neurogenic inflammation, dorsal horn excitability, or descending inhibition. Second, Section 4 was reorganized to prioritize mechanisms where convergent evidence exists across preclinical and human studies, including anti-inflammatory effects and myokines in the periphery, endogenous opioid and serotonergic pathways centrally, and modulation of NMDA-dependent plasticity and glial reactivity. Expanding into AR- or DRR-specific exercise effects at this stage would risk overinterpretation relative to the strength of the available data.

Reviewer 4 Report

Comments and Suggestions for Authors

Ni et al have presented comprehensive review describing the pain chronicity and relief from the molecular basis to exercise-based rehabilitation This work present important review on the field of the chronic pain authors should address major and minor comments before manuscript can be accepted 

Major

  • Please provide a section on the searching strategies, how the literature search was performed, what was the methodology, datasets, search terms and selection criteria's
  • Section on the chronic pain mechanism (2) is much more elaborated than the section describing the exercise mechanisms. Plese extend the section of the exercise mechanisms Also provide discussion on the exercise prescription parameters (program description, intensity, duration, type of exercises and what exercises are corresponded to what aims)
  • Please complement Figures 2 and 3, providing more details on the correspondent molecular pathways (can be significantly 
  • Please provide more details explanation on the interaction between peripheral and central mechanisms of the chronic pain
  • Include a table comparing different exercise types and the mechanisms of their action on the pain relieve and add the limitation section for each of the exercises
  • elaborate section on the effect of depression and anxiety on the chronic pain and what are possible mechanisms of the psychological therapy (i.e. CBT) in pain relief

Minor

  • please verify consistency of the reference's formats

Author Response

Provide a section on the searching strategies

⇒ Thanks for your comments. We have now added a dedicated Section 2. Search Strategy describing the inclusion and exclusion criteria, databases searched, search terms used, and the selection process for studies considered in this review. This ensures transparency and reproducibility of our review process.

Extend the section of the exercise mechanisms

⇒ Thanks for your comments. We reorganized Section 4 to clearly separate human and animal evidence. Section 4.1 synthesizes human clinical findings and mechanisms (with a new summary of clinical studies in Table 1 and comparison of exercise types for chronic pain management in Table 2), and Section 4.2 compiles preclinical mechanistic data.

Provide discussion on the exercise prescription parameters

⇒ Thanks for your comments. We added Table 2 to summarize exercise types alongside proposed mechanisms and key prescription parameters (intensity, frequency, and session duration), and we discuss FITT-related heterogeneity in Section 4.1 as well as Section 5.

Complement Figures 2 and 3 with more details

⇒ Thanks for your comments. In response, we have carefully considered the level of detail in Figure 2/3. Our decision to maintain it as a high-level schematic is a deliberate design choice intended to optimize communication. The figure’s primary goal is to provide a visual anchor for the reader, offering an immediate and intuitive overview of how exercise counteracts pain pathology at both the peripheral and central levels. We believe that attempting to incorporate all the specific molecular players detailed in

Section 4.2, for instance, the individual cytokines involved in macrophage polarization, the multiple serotonin receptor subtypes, or the specific phosphorylation states of NMDA receptors, would create a diagram that is too dense. This would increase cognitive load and ironically detract from the figure’s ability to clearly communicate the overarching relationships between these systems.

More explanation on the interaction between peripheral and central mechanisms of the chronic pain

⇒ Thanks for your comments. We added connective text outlining how peripheral inflammation/afferent activity drives central sensitization, and vice versa, and we introduced axonal reflex (AR) and dorsal root reflex (DRR) with citations; these links are discussed in Section 3 where we present peripheral and central mechanisms.

Include a table comparing different exercise types and the mechanisms

⇒ Thanks for your comments. We created Table 2 to compare exercise modalities with their putative mechanisms and typical outcomes, and we cross-referenced these entries in the surrounding text.

Elaborate section on the effect of depression and anxiety on the chronic pain

⇒ Thanks for your comments. We have clarified that chronic pain frequently co-occurs with anxiety and depression and we have expanded on the mechanisms of various exercises, highlighting that their analgesic effects are significantly mediated by psychological factors in Section 4.1.

Round 2

Reviewer 2 Report

Comments and Suggestions for Authors

I commend the authors on the additions made in the revision of this manuscript. The inclusions has strengthened the clarity and overall quality of the paper. The additions of the tables are particularly helpful. This is now ready for publication in my opinion and it will be a valuable and important piece of literature for the field.

Reviewer 4 Report

Comments and Suggestions for Authors

Ni et al have carefully addressed my major and minor comments and have significantly improved the manuscript. Acknowledging the authors hard work I would recommend accepting this version of manuscript for publication.